# Role of Creatine Supplementation in Conditions Involving Mitochondrial Dysfunction: A Narrative Review

**DOI:** 10.3390/nu14030529

**Published:** 2022-01-26

**Authors:** Robert Percy Marshall, Jan-Niklas Droste, Jürgen Giessing, Richard B. Kreider

**Affiliations:** 1Medical Department, RasenBallsport Leipzig GmbH, 04177 Leipzig, Germany; jan-niklas.droste@redbulls.com; 2Faculty of Natural and Environmental Sciences, Institute of Sports Science, Universität Koblenz-Landau, 76829 Landau, Germany; giessing@uni-landau.de; 3Exercise & Sport Nutrition Lab, Human Clinical Research Facility, Department of Health & Kinesiology, Texas A&M University, College Station, TX 77843, USA; rbkreider@tamu.edu

**Keywords:** mitochondriopathia, cardiac infarction, chronic fatigue syndrome, long COVID, ischemia, hypoxia, stroke, neurodegenerative diseases, oxidative stress, noncommunicable disease

## Abstract

Creatine monohydrate (CrM) is one of the most widely used nutritional supplements among active individuals and athletes to improve high-intensity exercise performance and training adaptations. However, research suggests that CrM supplementation may also serve as a therapeutic tool in the management of some chronic and traumatic diseases. Creatine supplementation has been reported to improve high-energy phosphate availability as well as have antioxidative, neuroprotective, anti-lactatic, and calcium-homoeostatic effects. These characteristics may have a direct impact on mitochondrion’s survival and health particularly during stressful conditions such as ischemia and injury. This narrative review discusses current scientific evidence for use or supplemental CrM as a therapeutic agent during conditions associated with mitochondrial dysfunction. Based on this analysis, it appears that CrM supplementation may have a role in improving cellular bioenergetics in several mitochondrial dysfunction-related diseases, ischemic conditions, and injury pathology and thereby could provide therapeutic benefit in the management of these conditions. However, larger clinical trials are needed to explore these potential therapeutic applications before definitive conclusions can be drawn.

## 1. Introduction

Creatine (*N*-aminoiminomethyl-*N*-methyl glycine) is a naturally occurring and nitrogen containing compound synthesized from the amino acids glycine, methionine that is classified within the family of guanidine phosphagens [1,2]. About one half the daily need for creatine is obtained from endogenous synthesis while the remaining is obtained from the diet, primarily red meat, fish, or dietary supplements [3,4]. Creatine is mainly stored in the muscle (95%) with the remaining found in the heart, brain, and testes [3,4,5,6], with about 2/3 in the form of PCr and the remaining as free creatine [4,5,7]. The metabolic basis of creatine in health and disease has been recently reviewed in detail by Bonilla and colleagues [1] (see Figure 1). Briefly, adenosine triphosphate (ATP) serves as the primary source of energy in most living cells. Enzymatic degradation of ATP into adenosine diphosphate (ADP) and inorganic phosphate (Pi) liberates free energy to fuel metabolic activity. However, only a small amount of ATP is stored in the cell. Energy derived from the degradation of phosphocreatine (PCr) serves to resynthesize ADP and Pi back to ATP to maintain cellular function until glycolysis in the cytosol and oxidative phosphorylation in the mitochondria can produce enough ATP to meet metabolic demands. Creatine also plays an important role in shuttling Pi from the mitochondria into the cytosol to form PCr to help maintain cellular bioenergetics (i.e., Creatine Phosphate Shuttle) [8]. In this way, PCr can donate its phosphate to ADP, thereby restoring ATP for cellular needs leaving creatine in the cytosol to diffuse back into the mitochondria to shuttle the next phosphate to locations far from its production site [8]. The ATP stored in the cells is usually sufficient for energy depletion that lasts less than two seconds. However, another two to seven seconds of muscle contractions are fueled by depleting available PCr stores [9]. Together, the ATP–PCr energy system provides energy to fuel short-term explosive exercise. Increasing PCr and creatine in muscle provides an energy reserve to meet anaerobic energy needs, thereby providing a critical source of energy particularly during ischemia, injury, and/or in response to impaired mitochondrial function [8,10].

Numerous studies over the last three decades have shown that creatine monohydrate (CrM) supplementation (e.g., 4 × 5 g/day for 5–7 days or 3–6 g/day for 4–12 weeks) increases muscle creatine and PCr content by 20–40% [5,11,12,13,14,15] and brain creatine content by 5–15% [16,17,18,19,20,21]. Creatine monohydrate supplementation has been reported to safely improve high-intensity exercise performance by 10–20% leading to greater training adaptations in adolescents [22,23,24,25,26], young adults [27,28,29,30,31,32,33,34,35,36,37,38], and older individuals [21,39,40,41,42,43,44,45,46,47,48]. No clinically significant side effects have been reported other than a desired weight gain [49]. Additionally, there is little to no evidence that CrM causes anecdotal reports of bloating, gastrointestinal distress, disproportionate increase in water retention, increased stress on the kidneys, increased susceptibility to injury, etc. [49,50]. In fact, studies directly assessing whether creatine causes some of those issues found no or opposite effects. As a result, there has been interest in assessing whether CrM supplementation may benefit a number of clinical populations including conditions that impair mitochondrial function [6]. The rationale is that since CrM supplementation can increase high-energy phosphate availability and also has antioxidant, neuroprotective, anti-lactatic, and calcium-homoeostatic effects, increasing phosphagen availability may help improve cell survival and/or health outcomes in conditions in which mitochondrial function is compromised (e.g., ischemia, injury, and/or non-communicable chronic diseases). The purpose of this review is to examine the literature related to the role of CrM supplementation in the management of various conditions characterized by mitochondrial dysfunction and make recommendations about further work needed in this area.

## 2. Methods

The methodological basis of this narrative review is a selective literature search in the PubMed database, supplemented by a free Internet search (German and English). In a first explorative step, the search terms “creatine supplementation” and/or “mitochondrial dysfunction” and “creatine” and/or “mitochondrial disease” were used. After a first analysis of the searched literature identifying 68 articles, a new selective literature search was performed in the sources described above using the terms mentioned above, adding relevant cited sources and cross-references. Subsequently, titles, abstracts and finally full-text articles were examined by the scientific team with regard to the suitability of the articles in terms of content and, in a subsequent step, in terms of quality. After the qualitative criteria had been verified, the content exploration was carried out following thematic questions related to the role of creatine in context: (1) Ergogenic role in mitochondrial dysfunction; (2) Noncommunicable chronic diseases (NCD); (3) Cardiovascular disease and ischemic heart failure; (4) Traumatic and ischemic CNS injuries; (5) Neurodegenerative disorders; (6) Psychological disorders; and (7) Chronic Fatigue Syndrome, Post Viral Fatigue Syndrome and Long COVID.

## 3. Creatine’s Ergogenic Role in Mitochondrial Dysfunction

Although there is not clear definition of mitochondrial dysfunction, it generally refers to conditions that reduce the ability of the mitochondria to contribute to production of energy in the form of ATP. However, any alteration of normal mitochondrial function could be called “mitochondrial dysfunction” as well [51]. Mitochondrial dysfunction can be of primary origin through inheriting pathological altered mitochondrial DNA (mtDNA) or acquiring secondary dysfunction through aging and exposure to mtDNA damaging processes [52,53]. This can be due to traumatic ischemic (blood deficient) or anoxic (oxygen deficient) as well as chronic conditions. Most common reasons for mitochondrial dysfunction are hypoxia, overexpression of reactive oxygen species (ROS), and an alteration of the intracellular calcium homoeostasis. Since creatine supplementation increases the availability of PCr, it may help cells withstand ischemic challenges and/or offset energy deficits associated with mitochondrial dysfunction

### 3.1. Acute, Traumatic Mitochondrial Dysfunction

Figure 2 shows the schematic sequence of an acute traumatic mitochondrial dysfunction with possible subsequent ischemia. The mechanical forces of injury result in an influx of calcium, potassium, and sodium. A calcium gradient is created, which reduces mitochondrial function [54,55]. In addition, an injury can lead to short-term ischemia (hypoxia) due to swelling, edema formation, development of neuroinflammation, obstruction of vessels, or hemorrhage [56]. The resulting oxygen deficiency interrupts the respiratory chain in the mitochondria. In both cases, the cell must switch to the energetic emergency plan and produce energy glycolytically, thereby increasing lactate production [57,58,59,60,61]. Oxygen radicals are generated, causing oxidative stress. This leads to cell damage and ultimately to cell death (apoptosis) [62,63,64]. If sufficient creatine phosphate reserves are present, the cell can compensate short-term energy deficits. ATP-dependent calcium transporters can counteract the calcium gradient under consumption of ATP and PCr, maintain the cell milieu, and thus normalize mitochondrial function [65,66]. Oxygen radicals can be intercepted [67]. Even transient hypoxia of a few seconds can be counteracted by the body in this way [68]. There is evidence that creatine and cyclocreatine inhibit the mitochondrial–creatine kinase–adenine nucleotide translocator (Mi-Cr-ANT) complex and the mitochondrial permeability transition that is associated with ischemic injury and apoptosis [69]. Additionally, creatine enhances the ability of Mi-CK to shuttle ADP for oxidative phosphorylation and PCr formation, thereby decreasing mitochondrial membrane and production of reactive oxygen species (ROS) [70]. Since impairment in cellular energy production and increased oxidative stress are common features in several neuromuscular degenerative diseases, creatine supplementation may provide some therapeutic benefit [69,70]. In support of this premise, Sakellaris et al. [71,72] reported that oral administered creatine can be used as an additional supplement in treatment of acute mitochondrial dysfunction after brain injury. These studies showed clear improvement in clinical outcomes of patients with additional creatine-supplementation in comparison to no creatine-intake. Table 1 shows the level of evidence in humans that creatine supplementation may have a positive effect on treatment outcomes in patients with traumatic brain injury.

### 3.2. Chronic, Atraumatic Mitochondrial Dysfunction

Many chronic diseases such as cancer and age-related pathological conditions have been related to an altered mitochondrial function [73,74,75,76,77,78,79,80,81,82,83,84,85,86,87,88,89,90,91,92,93,94,95,96,97,98,99,100,101]. Chronic mitochondrial dysfunction is usually caused by slow changes in mitochondrial homeostasis eventually leading to an increase in ROS/NOS, glycolysis, and hyper-acidosis. There are multiple factors that directly damage mitochondrial function (Figure 3). Hypoxia is a common factor in conditions such as solid tumor, ischemia, or inflammation that leads to a depletion of oxygen and eventually through production of ROS to an alteration of intracellular proteins, lipids and DNA [89]. On the other hand, research was able to prove that malignant cells tend to create energy under glycolytic conditions although sufficient oxygen is provided. This pathological mechanism is called “Warburg Effect” [102,103]. This leads to an increase in cell acidity and an increase in ROS with damaging of DNA. Other factors leading to chronic mitochondrial dysfunction are toxic metals or reactive nitrogen species (NOS) [104]. An increase in ingested carbohydrates bigger than the individual needs leads to hyperinsulinemia. As a chronic condition, this will lead to an increase in receptor for advanced glycation end products (RAGE). Thus, nitrosative stress increases, manipulating mitochondrial function [105,106,107,108,109]. Increasing stress will lead to an intracellular accumulation of ammonium [110,111,112], ROS [113], lactate [114], ultimately inhibiting the Krebs cycle and oxidative metabolism.

Typical factors that lead to a disturbance in the cellular respiration are hypoxia, inflammation, viruses, mutations, oncogenes, age, radiation, and carcinogens [115]. The ultimate, most common denominators are reactive species which damage mtDNA. As soon as cellular defense systems such as antioxidants, intracellular energetic buffer, and enzymatic reactions are worn down, chronic alteration of cellular organelles begins [116]. As mentioned above, it is hard to differentiate in chronic mitochondrial dysfunction whether pathological conditions lead to hypoxia that produces ROS/NOS which eventually harms mtDNA or whether an altered mtDNA leads to an overexpression of ROS/NOS damaging itself [117]. It is widely accepted, however, that this chronic status is a vicious circle leading to a lethal cellular condition harming the host.

Magnetic resonance spectroscopy (MRS) is an analytical tool that detects electromagnetical signals that are produced by the atomic nuclei within the molecules. Thus, it can be used to (non-invasively) measure concentrations for specific molecules in tissue. This technique has extensively been used in neurological research to identify phosphorus and proton metabolites in tissue in vivo [118,119,120,121]. Using this, research was able to prove mitochondrial dysfunction in patients with bipolar disorders. These patients also suffered from an impaired energy production [122], increased levels of lactate (hyperacidotic state) [123] and PCr concentration [114,124,125]. Therefore, it was assumed that creatine supplementation could improve clinical outcome in cases of mitochondrial dysfunction. Creatine is able to buffer lactate accumulation by reducing the need for glycolysis [126], reducing ROS [127] and restoring calcium homeostasis. Table 2 presents an overview of the level of evidence for creatine supplementation for chronic, atraumatic mitochondrial dysfunction.

## 4. Noncommunicable Chronic Diseases (NCD)

Modern ways of (unhealthy) living like over nutrition, exposure to toxic substances, and sedentarism combined with an individual’s genetic background led to the development of NCD [90]. Four disease clusters are associated with NCD such as cardiovascular diseases, cancers, chronic pulmonary diseases, and diabetes mellitus [129]. NCD are associated with low-grade inflammation and an increase in oxidative stress [130]. Through the past decades, they have become the biggest health threat of modern society [131,132,133]. Lately, there has been a link established between NCD and mitochondrial dysfunction. Reduced oxygen consumption rates have been shown in cardiovascular diseases such as hypertension and atherosclerosis. Additionally, they suffer from calcium overload due to mitochondrial calcium mishandling and ROS overproduction [134,135,136,137]. Obesity [138,139,140,141] as well as diabetes mellitus [142,143,144,145,146,147,148,149] are associated with an increased mitochondrial fragmentation rate, impaired ATP production, as well as ROS overproduction and calcium mishandling. In regards to creatine and its connection to mitochondrial dysfunction, reduced levels were detected in human myocytes in diabetes mellitus [150], obesity [151], and hypertension [152]. Not surprisingly, NCD are the most common factors contributing to the development of an acute ischemic heart attack or acute ischemic brain disease (Figure 4).

Table 3 shows some of the studies that have been conducted on creatine supplementation in noncommunicable chronic diseases. Creatine’s benefits in physical activity and thus counteracting NCD development have been widely explained [20,153,154,155,156,157,158,159,160,161,162,163]. There is, however, substantial evidence for the beneficial effects of supplementation even without combining it with sports. The sole intake of creatine has been able to significantly lower blood lipids such as cholesterol and triglycerides, slow down the development of fatty liver, and lower the HbA1C in human and animal studies, thus improving the clinical outcome and progression of the metabolic syndrome [164,165,166].

## 5. Cardiovascular Disease and Ischemic Heart Failure

Optimal replenishment of creatine reserves was able (in experimental studies) to slow down disease progression of the other above mentioned NCD and cardiomyopathy. Therefore, creatine supplementation has been identified to be of special therapeutic interest in treatment of cardiovascular diseases and their course [167,168]. The heart has its own four creatine kinase (CK) isozymes, proving the importance of ensuring filled energy depots [169]. A gradual reduction of myocardial total creatine content has been shown on chronic heart failure in human as well as animal studies [170,171,172,173]. The ratio of PCr/ATP has been defined to better judge myocardial creatine metabolism [174]. Low ratios have been positively correlated with low contractile function, more severe heart failure symptoms, and a higher risk of mortality [175,176,177].

Creatine supplementation in patients with chronic heart failure and similar animal studies have not shown any beneficial effect on clinical outcome, neither on myocardial creatine concentrations [178,179,180]. The transmembrane Creatine-Transporter (CrT) seems to be the limiting factor in this matter [181]. Question remains if other creatine-analogues that pass intracellular without the need of CrT might prove of better help in cardiovascular diseases. The energy deficiency resulting from local hypoxia during an ischemic heart attack leads to mitochondrial dysfunction, which in turn can have arrhythmogenic consequences and lead to sudden cardiac death [182,183,184]. Therefore, it is not surprising that creatine plays a critical role during a cardiac ischemic event [185,186]. First in vitro studies allow the hypothesis that saturation of myocardial creatine stores may lead to protection in the event of a transient ischemic attack [49]. In this context, in animal studies, filled ATP stores have a positive inotropic, apoptosis-protective effect and counteract a post-ischemic inflammatory cascade [187].

Intravenous in vivo administration of phosphocreatine was able to confer significant myocardial protection after bypass surgery [188], resulting in a reduction in the incidence of ventricular fibrillation and myocardial infarction as well as arrhythmias [189]. The newly developed special form of creatine, cyclo-creatine, deserves special attention. After an oral loading phase prior to elective cardiac interventions (PCI, ACVB, HTX), cyclo-creatine has a similar protective effect against lethal events [183,187,190,191]. However, large-scale human studies have yet to confirm the initial promising results. Table 4 summarizes the level of evidence available on the role of creatine in cardiovascular disease and ischemic heart failure [187,188,189,190,191].

## 6. Traumatic and Ischemic Central Nervous System Injuries

Mitochondrial function and ATP production are crucial for the neuronal survival and excitability [193]. At the same time, however, mitochondrial dysfunction leads to the overproduction of ROS and neuronal apoptosis which is closely related to neurodegenerative diseases and cerebral ischemia [193,194,195,196,197]. Whereas earlier research mainly focused on mitochondrial bioenergetic roles, new studies have shown the importance of apoptotic signaling, mitochondrial biogenesis, and mitophagy in the development of cerebrovascular disease and stroke. Mitochondrial health is therefore essential for neurological survival and rehabilitation [198,199]. Reperfusion injury is another acute complication feared by medical doctors involving mitochondria and clinical outcomes [200,201]. Following reperfusion of the injured brain tissue, excessive ROS and calcium produced under hypoxic conditions are washed in the body’s periphery, causing damage on cellular and molecular level [202]. Intracellular calcium deregulation enhances neuronal cell death after stroke, giving the stability of the mitochondrial (calcium) permeability transition pore (mPTP) a special predictive measure [203].

The acute protective effects of creatine on the central nervous system (CNS) have long been known. Similar to the effect in the myocardium, energy buffering for short-term hypoxic conditions can be achieved by saturating intracellular PCr. This may lead to protection against ischemia and cell death, as well as calcium gradients created by mechanical stimuli [204,205,206]. In animal experiments, researchers were able to show that idiopathically caused brain damage and spinal cord injuries developed to a lesser extent after creatine oral administration [207,208]. Creatine supplementation also had a positive effect on infarct sizes after insult in ischemic mouse models [209]. These results suggest that creatine administration may lead to preventive CNS protection against concussions, traumatic brain injury, spinal cord injury, and insults [210].

Adding to the above-mentioned protective effects of Creatine during a hypoxic situation, special advantages of creatine on the CNS have been proven. The term excitotoxicity describes the destruction of neuronal cells due to pathological activation of its excitatory receptors [202]. Research was able to show that excitatory amino acids, such as Glutamate, become more neurotoxic when the cell’s energy levels are reduced by hypoxia [211]. Activation of the glutamate NMDA receptor correlates with reduced ATP and PCr levels [212]. Creatine was able to protect animal brain tissues from the apoptotic effects of excitatory amino acids [213,214]. Lastly, it was shown that Creatine stabilizes mPTP in rodent studies, thus protecting brain tissue from apoptosis and cell death [67]. Table 5 presents a summary of the level of evidence related to creatine supplementation for traumatic and ischemic CNS injuries [205,206,207].

## 7. Neurodegenerative Disorders

Ageing has been defined as a “progressive accumulation of changes with time that are associated with or responsible for the ever-increasing susceptibility to disease and death” [215]. Brain tissue is due to its high-energy demands especially vulnerable to mitochondrial deficits, ROS, hypoxia, and energy depletion [216,217]. Although ROS are of special need to neurons and brain tissue needed for synaptic plasticity, learning and memory function, their overproduction is closely related to nitration of proteins, mtDNA impairment and the development of neurodegenerative diseases, ageing, and cognitive deficits [218,219,220]. Insulin resistance and diabetes mellitus deteriorate these conditions and accelerate cognitive decline as well as incidence of neurogenerative diseases [221,222,223]. RAGE and ammonium level up the documented damage to mitochondria, neuronal cells, and brain tissue [224,225,226]. Alzheimer’s disease has already been named “type 3 diabetes“ [227]. Pathologically altered mitochondria have been shown to be swollen, have altered membrane potential, and reductions of ATP levels [228]. Therefore, mitochondrial protection and reduction of oxidative stress have been suggested to be of high therapeutic importance for the treatment of neurodegenerative disorders [229]. Anti-inflammatory nutrition, caloric restriction, as well as the use of supplements have been discussed to be improve mitochondrial functioning and cognition [230,231,232,233]. Various studies have also shown that creatine supplementation has a positive effect on cognition and brain function [234,235]. The effect was greater the more the participant was exposed to external stressors (e.g., hypoxia, sleep deprivation, etc.) [45,205] or the more complex the tasks were performed [236]. In this context, intake led to a lower need for sleep, earlier wake-up times, and improved sleep behavior [237].

Neurodegenerative diseases are usually characterized by the destruction or dysfunction of neurons in a specific brain area. Depending on the affected brain area, course, and severity, the forms of the disease differ. These include Alzheimer’s disease (MA), amyotrophic lateral sclerosis (ALS), multiple sclerosis (MS), Huntington’s disease (MH), and Parkinson’s disease (MP). Impaired energy balance with mitochondrial dysfunction and oxidative stress are common to all diseases [238]. Similar findings have been made with intellectual disability-related diseases [239]. This bioenergetic deficit is thought to lead to apoptosis and necrosis and ultimately to neuronal degeneration [240]. Therefore, it is reasonable to assume that an improvement in mitochondrial health could enable a positive influence on the course of the disease. Table 6 provides a summary of the level of evidence related to the role of creatine supplementation for neurodegenerative disorders [45,234,236]. Initial studies suggest that creatine supplementation may be neuroprotective. For example, in 2013, Kley and coworkers [241] conducted a Cochrane review on the role of creatine monohydrate supplementation for treating muscle disorders. The researchers found sound evidence from randomized clinical trials that creatine supplementation increased strength and functional capacity in muscular dystrophy and idiopathic inflammatory myopathy while having no effect in patients with metabolic-related myopathies and McArdle disease. More long-term research is needed to evaluate the long-term effects of creatine in neurodegenerative diseases that impair muscle function.

## 8. Psychological Disorders

In the 1980s, a link was established between bioenergetic deficits and depression [190,242,243,244], bipolar disorders [114,245,246], and obsessive–compulsive disorders [247,248]. It is believed that there is an increase in energy demand with depletion of PCr stores at the onset of disease [124,249]. In clinical trials with depressed patients [250,251,252], a positive effect on subjective impairment after adjuvant creatine supplementation could be demonstrated. The higher the increase in cerebral PCr after creatine supplementation, the lower the depressive or manic symptoms [253]. The combination of antidepressants and creatine was more effective than simple pharmacological medication [254]. Creatine administration was even effective when drug therapy with SSRIs proved to be ineffective [255]. In this content, creatine has also been discussed as a potential therapeutic agent in the treatment of drug addiction and its psychic related disorders [256]. Positive effects of creatine supplementation have also been reported in post-traumatic stress disorders [257]. Schizophrenic and stress patients seem to gain no benefit from creatine intake. There is, however, ongoing debate on higher dosage for a needed benefit in these sub-groups [258]. Table 7 presents a summary of the literature related to the effects of creatine supplementation on individuals with psychological disorders [251,252,255].

## 9. Chronic Fatigue Syndrome, Post Viral Fatigue Syndrome, and Long COVID

Fatigue is the most characteristic symptom of an energy deficit. There does not, however, exist a proper definition of the fatigue syndrome [259]. Fibromyalgia is a similar pathological entity closely related to CFS. Initially thought to be purely a psychological problem, linking fatigue to depression or other psychiatric diseases, newer research has been able to prove a metabolic dysfunction causing the symptoms [99,260,261]. Linking this clinical state with mitochondrial dysfunction was first able when lowered mitochondrial ATP levels were shown using MRS on patients with fatigue syndrome [262]. Later muscle biopsies and serum biomarkers have been able to show reduced mitochondrial biomarkers [263,264]. These markers have been Carnitine and CoQ10 [265]. On a mitochondrial level fatty acid metabolism was altered, electron transport chain was disrupted, there was a greater need in glucose concentrations and higher levels of lactate were shown [266]. Higher creatinine excretion via urine was shown to correlate positively with fatigue and pain severity. Being the end product of creatine, this urine marker could imply a higher turnover and depletion of the body’s creatine storage [267]. More recent hypotheses state that these alterations have been caused by an activation of immune–inflammatory pathways due to viral infections (e.g., Epstein Barr, Q Fever, Ross River Infection) [268].

Long COVID is a persistent fatigue state after Sars-2-CoV-2 infection [269,270]. Interestingly, even asymptomatic patients exhibited raised biomarkers involved in inflammation and stress response [271]. Long COVID, Chronic Fatigue Syndrome, and Post Viral Fatigue Syndrome are believed to be the same entity [248,272]. Supplementation of guadinioacteic acid, a precursor of creatine, was able to attenuate several aspects of fatigue in fibromyalgia patients [273]. In combination of experimental findings as well as these first promising clinical outcomes, creatine might be an important key in the rehabilitation process of CFS and Long COVID patients [274]. Table 8 summarizes the available literature on the effects of the creatine precursor GAA on chronic fatigue and Post-COVID syndrome [274].

## 10. Conclusions

This review summarizes creatine’s impact on mitochondrial function besides restoring ATP-storage. Creatine monohydrate is one of the best-known nutrient supplements mainly being used for improvement of athletic performance. However, there is growing evidence for a broader therapeutic spectrum of this nitrogen–amino-compound. Various health-promoting effects on cell-metabolism after the intake of creatine have been shown. Its impact on mitochondrial integrity has become of special interest. Mitochondrial dysfunction has become a central pathological hallmark of non-communicable diseases. The supplementation of creatine monohydrate may have some synergistic effects in the treatment of CND. This seems to be directly related to its protective effects on mitochondria. Different from pharmaceutical products, the intake of creatine is safe age- and gender-independent with nearly no side-effects [49,50]. Although these findings are promising, much of the available data has been generated with in vitro or animal studies. Therefore, there is a need to conduct more clinical trials in humans to assess the potential therapeutic effects of creatine monohydrate supplementation on conditions influencing mitochondrial function.

## Figures and Tables

**Figure 1 nutrients-14-00529-f001:**
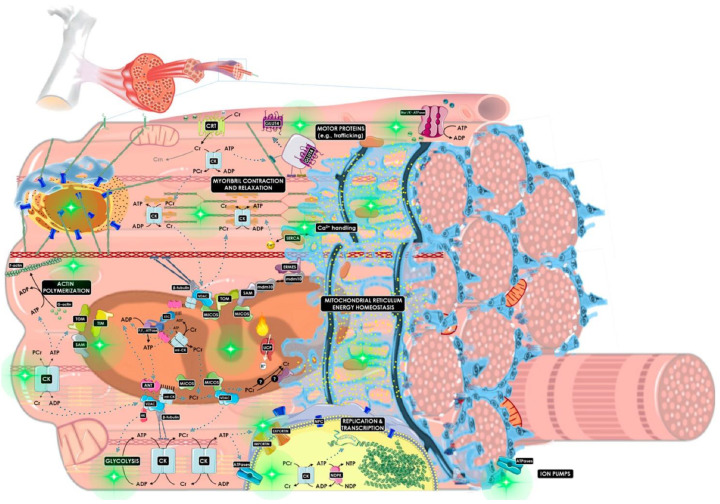
General overview of the metabolic role of creatine in the creatine kinase/phosphocreatine (CK/PCr) system [1]. The diagram depicts connected subcellular energy production and cellular mechanics of creatine metabolism. This chemo-mechanical energy transduction network involves structural and functional coupling of the mitochondrial reticulum (mitochondrial interactosome and oxidative metabolism), phosphagen and glycolytic system (extramitochondrial ATP production), the linker of nucleoskeleton and cytoskeleton complex (nesprins interaction with microtubules, actin polymerization, β-tubulins), motor proteins (e.g., myofibrillar ATPase machinery, vesicles transport), and ion pumps (e.g., SERCA, Na^+^/K^+^-ATPase). The cardiolipin-rich domain is represented by parallel black lines. Green sparkled circles represent the subcellular processes where the CK/PCr system is important for functionality. Several proteins of the endoplasmic reticulum–mitochondria organizing network (ERMIONE), the SERCA complex, the TIM/TOM complex, the MICOS complex, the linker of nucleoskeleton and cytoskeleton complex, and the architecture of sarcomere and cytoskeleton are not depicted for readability. ANT: adenine nucleotide translocase; CK: creatine kinase; Cr: creatine; Crn: creatinine; CRT: Na^+^/Cl^−^-dependent creatine transporter; ERMES: endoplasmic reticulum-mitochondria encounter structure; ETC: electron transport chain; GLUT-4: glucose transporter type 4; HK: hexokinase; mdm10: mitochondrial distribution and morphology protein 10; MICOS: mitochondrial contact site and cristae organizing system; NDPK: nucleoside-diphosphate kinase; NPC: nuclear pore complex; PCr: phosphocreatine; SAM: sorting and assembly machinery; SERCA: Sarco/Endoplasmic Reticulum Ca^2+^ ATPase; TIM: translocase of the inner membrane complex; TOM: translocase of the outer membrane complex; UCP: uncoupling protein; VDAC: voltage-dependent anion channel. Reprinted with permission. See Bonilla et al. [1] for more details about the metabolic basis of creatine in energy production and disease.

**Figure 2 nutrients-14-00529-f002:**
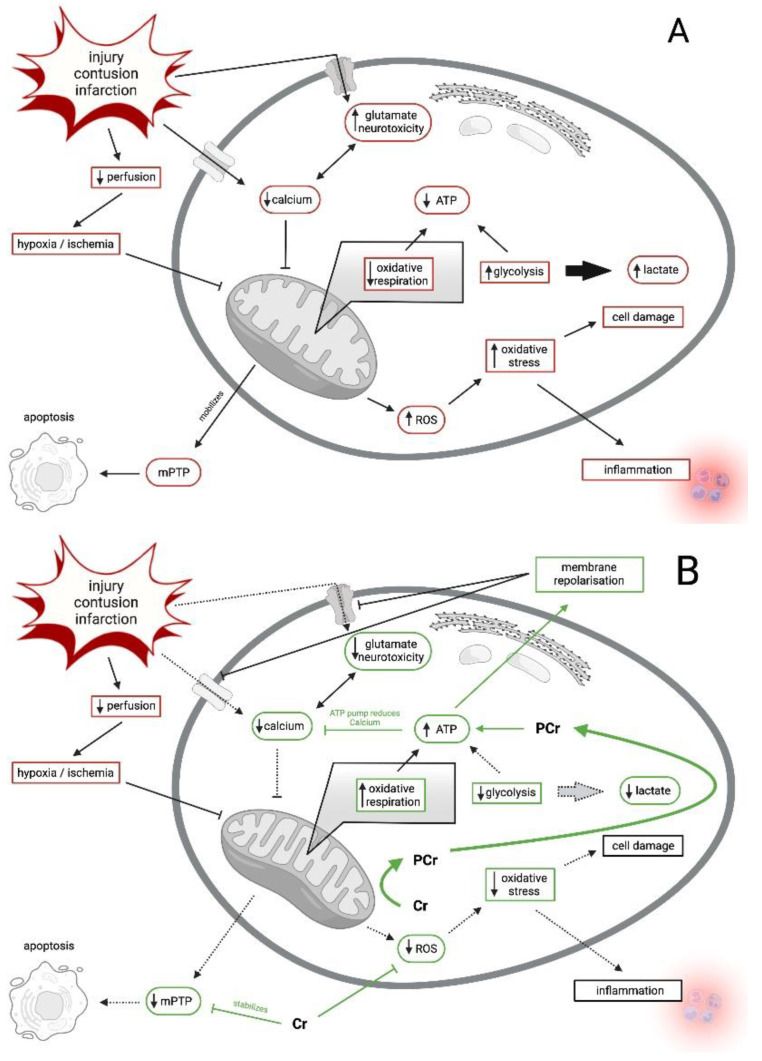
Panel **A**: Intracellular cascade after injury, infarction or contusion leads to mitochondrial dysfunction. Panel **B**: Impact of creatine on mitochondrial dysfunction. Green shows direct increase/stimulation of Cr/PCr, red shows direct decrease/inhibition of Cr/PCr, dotted line represents indirect impact of Cr/PCr on cellular pathways. ATP is adenosine triphosphate; Cr is creatine; PCr is phosphocreatine; ROS is reactive oxygen species; mPTP is mitochondrial permeability transition pore. Adapted from Dean et al. [55].

**Figure 3 nutrients-14-00529-f003:**
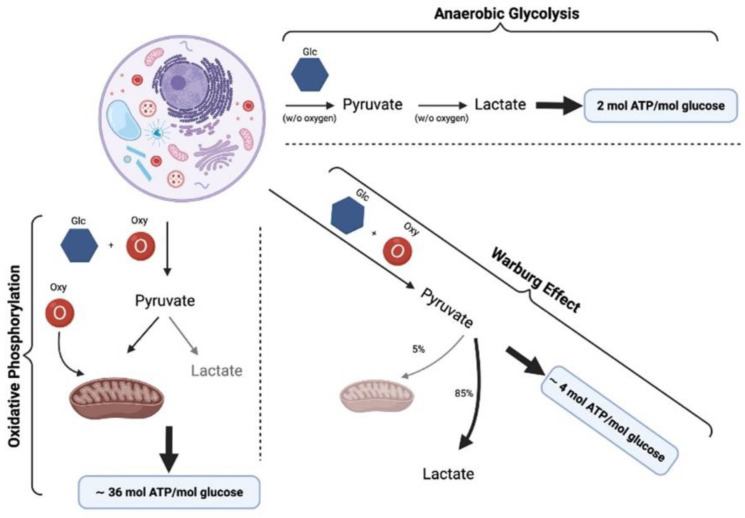
Warburg Effect: glycolysis produces 2 ATP instead of 36 ATP, in pathological tissues even despite aerobic conditions. Glc is glucose, Oxy is oxygen, ATP is adenosine triphosphate. Adapted from Vander Heiden et al. [91].

**Figure 4 nutrients-14-00529-f004:**
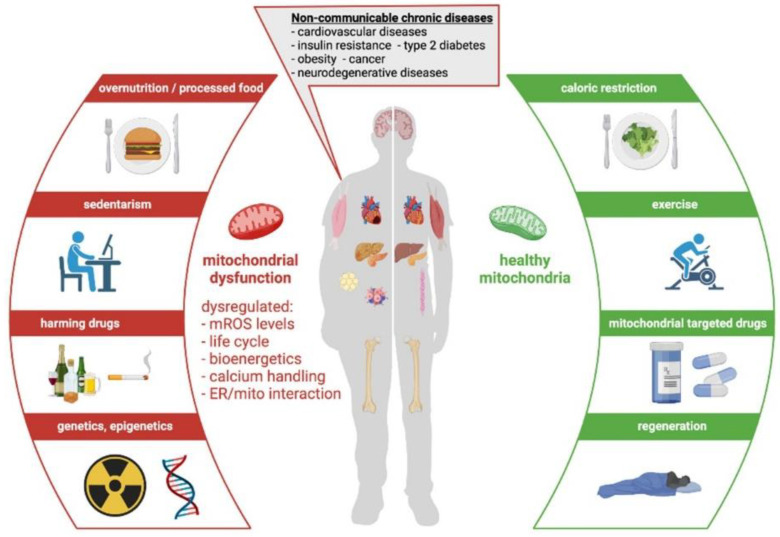
Mitochondrial dysfunction and non-communicable diseases. Adapted from Diaz-Vegas et al. [90].

**Table 1 nutrients-14-00529-t001:** Level of evidence for creatine supplementation in acute traumatic mitochondrial dysfunction.

Study	Disease	Subject	Treatment	Randomized	Subjects	Efficacy	Effect Role
Sakellaris et al. [71]	Traumatic brain injury	Human	0.4 g/kg per day for 6 months	Yes	39	Improved self-care, cognition, behavior functions and communication	Direct effect on disease
Sakellaris et al. [72]	Traumatic brain injury	Human	0.4 g/kg per day for 6 months	Yes	39	Reduced fatigue, headache and dizziness	Direct effect on disease

**Table 2 nutrients-14-00529-t002:** Level of evidence for creatine supplementation for chronic, atraumatic mitochondrial dysfunction.

Study	Disease	Subject	Treatment	Randomized	Subjects	Efficacy	Effect Role
Guimarães-Ferreira et al. [128]	-	Animal/vitro	5 g/kg per day for 6 days	no	39	Decrease in ROS in muscle tissue	Anima model
Kato et al. [124]	Bipolar disorder	Humans	None	No	25 (disease) vs. 21 (control)	Abnormal energy phosphate metabolism in bipolar disorder	No intervention, only descriptive, observational findings

**Table 3 nutrients-14-00529-t003:** Level of evidence of creatine’s role in noncommunicable chronic disease.

Study	Disease	Subject	Treatment	Randomized	Subjects	Efficacy	Effect Role
Rider et al. [151]	Obesity	Human	None	None	64	Deranged cardiac energetics and diastolic dysfunction in obesity group	Observational, disease related changes in metabolism
Scheuermann-Freestone et al. [150]	Diabetes Type 2	Human	None	None	36	Impaired myocardial and skeletal muscle metabolism (reduced PCR/ATP ratio)	Observational disease related changes in metabolism
Lamb et al. [152]	Hypertension	Human	None	None	24	Altered high-energy phosphate metabolism in hypertension. Cardiac dysfunction correlates with metabolic alterations	Observational, disease related changes in metabolism
Gualano et al. [164]	Diabetes Type 2	Human	5 g creatine for 12 weeks + physical activity program	Yes	25	Improved glycemic control in supplementation group (by GLUT-4 recruitment)	Direct effect on disease related metabolic effects
Earnest et al. [165]	Hyper-cholester-inaemia	Human	4 × 5 g creatine for 5 days and afterwards 2 times per day for 51 days (orally)	Yes	34	Minor reduction of total cholesterol during supplementation. Reduction of triacylglycerol’s and very-low-density-lipoprotein c 4 weeks after finishing supplementation	Direct effect of supplementation on metabolism.
Deminice et al. [166]	Fatty liver	Animal	Control vs. 0.25% choline diet vs. 0.25% choline + 2% creatine diet	None	24	Prevention of fat liver accumulation and hepatic events in creatine-fed group	Animal model

**Table 4 nutrients-14-00529-t004:** Level of evidence for creatine supplementation for chronic, atraumatic mitochondrial dysfunction.

Study	Disease	Subject	Treatment	Randomized	Subjects	Efficacy	Effect Role
Elgebaly et al. [187]	-	Animal/vitro	500 mg/kg BW	no	6	Better aortic flow, coronary flow, cardiac output, stroke volume, and stroke work	Animal model
Cisowski et al. [188]	Cardiac surgery	Humans	6 g 3 days pre-surgery, intra-surgical and two days post- surgery i.v.	yes	40	Reduced arrhythmic events, reduced need of ionotropic medication	Direct effect on surgical procedure
Ruda et al. [189]	Ischemic myocardial infarct	human	2 g bolus + 4 g/h over 2 h	Yes	60	Reduced arrhythmic events	Direct effect on short term outcome
Chida et al. [192]	Dilated Cardio-myopathy	Human	None	None	13	Plasma BNP level was correlated negatively with the myocardial phosphocreatine/adenosine triphosphate	Observational finding
Roberts et al. [191].	None	Animal	Oral creatine-feeding	None	Not clear	Higher cellular ATP during ischemia in creatine-fed rat hearts	Animal model

**Table 5 nutrients-14-00529-t005:** Level of evidence for the role of creatine supplementation in individuals with traumatic and ischemic CNS injuries.

Study	Disease	Subject	Treatment	Randomized	Subjects	Efficacy	Effect Role
Zhu et al. [206]	None/induced ischemia	Animal	2% creatine-supplemented diet for 4 weeks	None	6 per group	Reduction in ischemia induced infarct size	Animal model
Turner et al. [205]	None/induced hypoxia	Human	7-ds oral creatine-supplementation	Yes	15	Less decrease in cognitive performance, attentional capacity, corticomotor excitability for creatine-group	Human brain metabolism
Hausmann et al. [207]	None/induced spinal cord injury	Animal	4 weeks oral creatine-supplementation	none	20	Better locomotor scores after 1 week for creatine-group. Less scar tissue for creatine-group after 2 weeks	Animal model
Sullivan et al. [208]	None/induced traumatic brain injury	Animal	Mice: 0.1 mL/10 g/BW creatine monohydrate injection for 1, 3 or 5 days	none	40 mice/24 rats	Reduction of brain tissue damage size by 36% mice and 50% rats	Animal model
			Rats: 1% creatine diet for 4 weeks.				
Prass et al. [209]	None/induced experimental stroke	Animal	Creatine-rich diet (0%, 0.5%, 1%, 2% for 3 weeks	None	Unclear	Reduction of infarct size by 40% in 2% creatine-fed group	Animal model

**Table 6 nutrients-14-00529-t006:** Level of evidence for the role of creatine supplementation in individuals with neurodegenerative disorders.

Study	Disease	Subject	Treatment	Randomized	Subjects	Efficacy	Effect Role
Hammett et al. [234]	None	Human	20 g/d creatine for 5 days + 5 g/d for 2-days	Yes	22	Reduction of stress related blood oxygen level dependent in fMRI in creatine-group	Human metabolic response
Watanabe et al. [235]	None	Human	8 g/d for 5-days	Yes	24	Reduction of mental fatigue and increased brain oxygen consumption in creatine-group	Human metabolic response
McMorris et al. [236]	None	Human	4 × 5 g/d	yes	20	Better in central complex executive tasks with creatine while sleep deprivation	Human metabolic response
McMorris et al. [45]	None	Human	4 × 5 g/d	Yes	15	random number generation, forward number and spatial recall, and long-term memory	Human metabolism

**Table 7 nutrients-14-00529-t007:** Level of evidence for the role of creatine supplementation in individuals with psychological disorders.

Study	Disease	Subject	Treatment	Randomized	Subjects	Efficacy	Effect Role
Kondo et al. [250]	Adolescent major depressive disorder	Human	4 g/d creatine for 8 weeks	None	15	Reduction in children-depression symptom scores. Significant increase in brain phosphocreatine level.	Direct effect on disease (no RCT)
Roitman et al. [251]	Treatment resistant depression	Human	3–5 g/d creatine for 4 weeks	None	8 unipolar depressed patients and two bipolar patients	Development of hypomania/mania in bipolar patients. Improved Hamilton Depression Rating Scale, Hamilton Anxiety Scale, and Clinical Global Impression for 7 of 8 unipolar depressed patients	Direct effect on disease (no RCT)
Toniolo et al. [252]	Depressive episode of Bipolar Type 1 and Type 2	Human	6 g/d creatine for 6 weeks	Yes	35	No significant difference in Montgomery-Åsberg Depression Rating Scale by intervention but higher remission rate in creatine supplemented group	Direct effect on disease
Kondo et al. [255]	Adolescent with SSRI resistant major depressive disorder	Human	0 g vs. 2 g vs. 4 g vs. 10 g creatine supplementation for 8 weeks	Yes	34	Clinical depression scores correlated inversely with brain phosphocreatine (PCR) levels. PCR level improved with higher dose.	Potential direct effect on disease

**Table 8 nutrients-14-00529-t008:** Summary of literature on the effects of creatine precursors on chronic fatigue and Post-COVID syndrome.

Study	Disease	Subject	Treatment	Randomized	Subjects	Efficacy	Effect Role
Ostojic et al. [264]	Chronic Fatigue syndrome	Human	2 g, 4 g oral Guanidinoacetic Acid for 3 months vs. placebo	Yes	21	Higher muscle creatine-phosphate level and better oxidative capacity. However, no significant improvement of fatigue symptoms	Direct effect on disease related metabolism

## Data Availability

Not applicable.

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
