# Peer review of "Role of Creatine Supplementation in Conditions Involving Mitochondrial Dysfunction: A Narrative Review"

_nutrients, 2022, doi:10.3390/nu14030529_

Round 1

Reviewer 1 Report

The study is well done, the material is large enough and the methods look reliable. However the study is based on extensive and very recent literature, gives some new information.

Author Response

Response to Reviewer #1

Comment

The study is well done, the material is large enough and the methods look reliable. However, the study is based on extensive and very recent literature, gives some new information.

Response

Thank you. This paper is part of invited papers for the Special Issue on Creatine for Health and Clinical Diseases.  Although some of the reviews published in this special issue mention some diseases affected by mitochondrial disease, the editors want to focus a paper on this issue specifically.  We feel the paper, with revisions, will be an important contribution to the scientific literature and increase awareness and prompt others to focus more research on this area.

Reviewer 2 Report

The manuscript is a review “aimed at having a deeper look into creatine’s possible benefits in the treatment of mitochondrial dysfunction“.

General comments:

The clinical scope and specific aims should be redefined, introducing the text body. In this work,  the objectives are vague and poorly mentioned by authors (lines 108-110), whereas the criteria and content of “thematic questions” remain elusive (lines 90-95).

The use of creatine supplementation in Health and Disease has been the object of comprehensive reviews during the last decade. The title of the present work suggests a potential approach focused on the molecular effects of creatine on cellular bioenergetics. However, this is a very superficial review concerning the mechanistic bases of mitochondrial dysfunction in disease pathophysiology and the potential benefits of creatine. In biochemical terms, numerous inaccuracies are encountered along with text, intersecting complex multifactorial pathologies. No clear mechanisms are reviewed within the present perspective, underlying the pleiotropic implications of mitochondrial dysfunction and potential creatine-mediated role, which in my view is not desirable. All figures within the manuscript are exact copies from other references and thus, original schemes should be proposed.

The final sentences of the abstract underline the biased perspective of the use of CrM supplementation as a panacea for treating numerous pathologies. These ideas are not in accordance with the final sentences of the “Summary” (item 10) which I suggest renaming as “Final remarks or Conclusions” where a certain caution is highlighted by authors: “Data has often been generated with in-vitro or animal studies. There is still need for larger clinical studies to be able to make final statements on effect and intake of creatine as a therapeutic agent”. Although the topic is potentially interesting, this reviewer does not find enough consistency in the report of studies details, models, or clinical studies. The potential adverse effects of creatine supplementation namely on renal function are not properly reviewed or even mentioned.

Author Response

Response to Reviewer #2

Comment

The manuscript is a review “aimed at having a deeper look into creatine’s possible benefits in the treatment of mitochondrial dysfunction“.

General comments:

The clinical scope and specific aims should be redefined, introducing the text body. In this work, the objectives are vague and poorly mentioned by authors (lines 108-110), whereas the criteria and content of “thematic questions” remain elusive (lines 90-95).

Response

Thank you. This paper is one of about 20 invited papers for the Special Issue on Creatine for Health and Clinical Diseases.  Although some of the reviews published so far in this special issue mention some diseases affected by mitochondrial dysfunction, the editors wanted to focus a paper on this issue specifically.  We believe we have provided a detailed overview of the role of creatine supplementation in mitochondrial dysfunction and that this paper complements other reviews published in this special issue to date. We feel the paper, with revisions, will be an important contribution to the scientific literature and increase awareness, and prompt others to focus more research on this area. We thank the reviewer for offering constructive comments and have made the following changes:

  1. We revised the title to “Role of creatine supplementation in conditions involving mitochondrial dysfunction: - a narrative review”
  2. We revised the introduction to provide more clarity about the purpose and intent of the review.
  3. We developed original artwork for figures presented.
  4. We highlighted that larger clinical trials are necessary to examine the potential therapeutic role of creatine in conditions involving mitochondrial dysfunction.

Comment

The use of creatine supplementation in Health and Disease has been the object of comprehensive reviews during the last decade. The title of the present work suggests a potential approach focused on the molecular effects of creatine on cellular bioenergetics. However, this is a very superficial review concerning the mechanistic bases of mitochondrial dysfunction in disease pathophysiology and the potential benefits of creatine. In biochemical terms, numerous inaccuracies are encountered along with text, intersecting complex multifactorial pathologies. No clear mechanisms are reviewed within the present perspective, underlying the pleiotropic implications of mitochondrial dysfunction and potential creatine-mediated role, which in my view is not desirable. All figures within the manuscript are exact copies from other references and thus, original schemes should be proposed.

Response

Because this paper is one of about 20 invited papers for the Special Issue on Creatine for Health and Clinical Diseases, including a paper by Bonilla et al. Metabolic Basis of Creatine in Health and Disease: A Bioinformatics-Assisted Review. Nutrients. 2021;13(4):1238. Published 2021 Apr 9. doi:10.3390/nu13041238; we did not ask authors to re-write the metabolic basis for each article submitted in this special edition.  However, we revised this section in the introduction, referred readers to the more thorough review, and included one figure from that review that overviews the metabolic role creatine plays in metabolism thereby providing a basis how creatine may impact health and disease.  We believe these changes addresses your comments.

Comment

The final sentences of the abstract underline the biased perspective of the use of CrM supplementation as a panacea for treating numerous pathologies. These ideas are not in accordance with the final sentences of the “Summary” (item 10) which I suggest renaming as “Final remarks or Conclusions” where a certain caution is highlighted by authors: “Data has often been generated with in-vitro or animal studies. There is still need for larger clinical studies to be able to make final statements on effect and intake of creatine as a therapeutic agent”. Although the topic is potentially interesting, this reviewer does not find enough consistency in the report of studies details, models, or clinical studies. The potential adverse effects of creatine supplementation namely on renal function are not properly reviewed or even mentioned.

Response

Thank you. We have revised the abstract to be consistent with the conclusion.  Although there are a number of potential health and clinical related benefits from creatine supplementation, the purpose of this special issue is to prompt more research in this area, including on the role of creatine in conditions where mitochondrial energy production may be limited. 

In terms of side effects, creatine has been FDA reviewed and has generally recognized as safe (GRAS) status for use as a dietary ingredient in supplements and foods. The European Union Commission, Health Canada, and Therapeutic Goods Administration in the Department of Health of Australia also consider pure creatine monohydrate as safe with authorization to include CrM in dietary supplements and in food. The only side effect reported from hundreds of controlled clinical trials has been weight gain and occasional reports of gastrointestinal issues (that are typically similar to placebo). A number of clinical trials have been conducted to determine whether anecdotally reported side effects and/or isolated case study reports about adverse events have any merit.  This includes several papers on the acute and long-term effects on kidney function. Nevertheless, given your comments, we added citations of several reviews and position stands that have concluded that creatine monohydrate supplementation is safe and does not cause untoward side effects in healthy or clinical populations. Given the widespread, worldwide use of creatine monohydrate over the last 30 years, there does not appear to be any side effects. Conversely, as highlighted in the over 20 papers in this special issue, there are a number of potential health and therapeutic benefits. This issue, and this paper, is designed to increase awareness about the science related to the potential health benefits of creatine supplementation in hopes that more scientists and clinicians further evaluate these potential applications and the public can be well-informed about creatine supplementation.

Once again, we thank you for your comments and believe they have improved this paper.